# Mechanism of SARS-CoV-2 resistance to nucleotide analog-based antivirals

Chang Liu [1] ✉, Yu Li[2], Xiaocong Cao[1], Ryan J. Gleason[1], Bin Liu [3] ✉ & Yang Yang [2] ✉

The remarkable ability of SARS-CoV-2 to resist many nucleotide analog (NA)-based antivirals represents a formidable challenge to therapeutic efforts. Here, we reveal fundamental insights into how its unique proofreading exoribonuclease (ExoN) counteracts two representative NA antivirals, bemnifosbuvir and sofosbuvir, which are designed to inhibit the viral RNA polymerase (RdRp). Our findings unveil that NA incorporation alters RNA-binding dynamics, significantly increasing the affinity of RNA to ExoN while weakening its interaction with RdRp. This shift likely facilitates RNA dissociation from RdRp, subsequent recognition by ExoN, and excision of NAs. Strikingly, we elucidate the mechanism underlying varied levels of resilience of different NAs to ExoN excision. Our cryo-EM structures of ExoN in complex with either of the two NA-incorporated RNAs reveal previously unknown ExoN-NA interactions mediated by the functional groups on the modified ribose rings of NAs, illuminating the key determinants of their recognition and excision. Furthermore, we identify an allosteric regulatory loop of ExoN that promotes the full activation of ExoN but is displaced by the binding of NAs exhibiting resilience to ExoN excision. These discoveries provide a molecular framework for understanding SARS-CoV-2 resistance to NA-based antivirals and highlight mechanisms that could be exploited to improve anti-coronavirus drug design.

The COVID-19 pandemic highlights the significance of studying the mechanisms of coronavirus genome replication and transcription, which constitute a major part of the viral life cycle, to potentiate our ability to combat these deadly human pathogens. Coronaviruses are enveloped positive-sense RNA viruses with usually large genomes (~30 kilobases)[1], the replication and transcription of which are carried out by a multi-subunit viral replication and transcription complex (RTC)[2–5]. The viral RTC comprises an array of non-structural proteins (nsps) that possess various RNA synthesis and processing activities essential for the virus life cycle[2–4], hence representing promising targets for new anti-coronavirus treatments[3,6–8]. Among them, nsp14, together with its stimulating nsp10, functions as an exoribonuclease complex (hereafter referred to as ExoN) that proofreads RNA synthesis and corrects mismatches[9,10] introduced by the error-prone viral RNA-dependent RNA polymerase (RdRp)[11,12], thus improving the fidelity of viral genome replication and transcription. In the RNA virus world, proofreading ExoN is found only in coronaviruses and a few closely related virus families with large RNA genomes[13]. Beyond its role in removing mis-incorporated nucleotides during viral RNA synthesis, this unique proofreading ExoN also has an unfortunate side effect from a therapeutic point of view. It "erases" many nucleoside/tide analog (NA)-based antivirals, such as sofosbuvir and ribavirin, after them being incorporated into viral RNAs[3,10,14–16], thus challenging the effectiveness of this class of antivirals that could otherwise interrupt viral RNA synthesis carried out by the viral RdRp.

[1]Department of Biophysics and Biophysical Chemistry, The Johns Hopkins University School of Medicine, Baltimore, MD, USA. [2]Roy J. Carver Department of Biochemistry, Biophysics and Molecular Biology, Iowa State University, Ames, IA, USA. [3]Section of Transcription and Gene Regulation, The Hormel Institute, University of Minnesota, Austin, MN, USA. ✉e-mail: cliu207@jhmi.edu; liu00794@umn.edu; yan9yang@iastate.edu

NA-based antivirals have been widely used to treat diseases caused by many RNA viruses, including hepatitis C virus (HCV) and Ebola virus[17–21]. The antiviral effectiveness of a wide range of NA-based antivirals depends on their incorporation into synthesizing viral RNA chains by the error-prone viral RdRp after being metabolized from their prodrug forms to the active triphosphate forms by cellular enzymes. Several NA-based antivirals, such as remdesivir, bemnifosbuvir (also known as AT-527), and sofosbuvir, have been shown to inhibit SARS-CoV-2 RdRp activity in vitro and, to some extent, viral replication in cells[22–29]. These nucleotide analogs are incorporated into the viral RNAs and act as chain terminators to inhibit RdRp and thus stall RNA synthesis[23,27,29]. Another type of NA-based antivirals, including molnupiravir[30–32], favipiravir[18,19,33], and ribavirin[17,34,35], act by inducing lethal mutagenesis rather than chain termination due to their promiscuous base pairing properties.

Multiple lines of evidence suggest that the effectiveness of many NA-based antivirals in inhibiting the replication and infection of coronaviruses is significantly undermined by the coronavirus proofreading ExoN. First, double-stranded RNA (dsRNA) substrates bearing the monophosphate form of different NA-based antivirals, including remdesivir, sofosbuvir, bemnifosbuvir, and ribavirin, at the 3′ end are readily digested by SARS-CoV-2 ExoN[14,29,36–38]. Second, ExoN inactivation has been shown to markedly increase the sensitivity of different coronaviruses to various NA-based antivirals[16,39], suggesting that ExoN is responsible for removing these NA-based antivirals misincorporated by RdRp from viral RNAs. However, it is unclear how the incorporation of NA-based antivirals into the viral RNA affects the dynamics of RNA binding by ExoN and RdRp, which seemingly compete for the same 3′ end of dsRNA substrates. In addition, the molecular details regarding how ExoN recognizes these different types of NAs and how various types of modifications on the NAs affect their excisions by ExoN are completely lacking. Such information is critical

for developing next-generation antivirals that can evade the coronavirus ExoN-mediated proofreading mechanism.

Here, we determined the cryogenic-electron microscopy (cryo-EM) structures of SARS-CoV-2 ExoN in complex with the monophosphate form of two clinically important NA-based antivirals (bemnifosbuvir or sofosbuvir) incorporated at the 3′-end of a dsRNA substrate at a resolution of 2.4–2.9 Å (Supplementary Figs. 1–4 and Supplementary Table 1). These high-resolution cryo-EM structures offer unprecedented insights into proofreading ExoN-mediated coronavirus resistance to NA-based antivirals. Together with extensive biochemical characterizations, we reveal that the incorporation of either of the two NAs increases the binding affinities of the RNAs to SARS-CoV-2 ExoN while weakening their interactions with SARS-CoV-2 RdRp. Such reciprocal changes in RNA-binding by ExoN and RdRp likely induce the dissociation of the RNAs from RdRp and facilitate the subsequent recognition of the RNAs, excision of the NAs, and rescue of NA-stalled RNA synthesis by ExoN. In addition, our findings unveil the profound impacts of ribose modifications and nucleobase identities on the efficiency of NA excision by ExoN and thus cast fresh light on the strategies for overcoming viral proofreading and improving therapeutic interventions against SARS-CoV-2 and other coronaviruses.

## Results

### Incorporation of bemnifosbuvir or sofosbuvir weakens RdRp-RNA binding

The active metabolized forms of bemnifosbuvir and sofosbuvir are analogs of guanosine 5′-triphosphate (GTP) and uridine 5′-triphosphate (UTP), respectively (Fig. 1a, b). Both bemnifosbuvir and sofosbuvir contain a 2′-deoxy-2′-fluoro-2′-C-methyl-modified ribose ring (Fig. 1a, b), which prevents correct alignment of the incoming NTP, thus terminating RNA synthesis[29]. Bemnifosbuvir and sofosbuvir were originally developed for the treatment of HCV, but lately have also

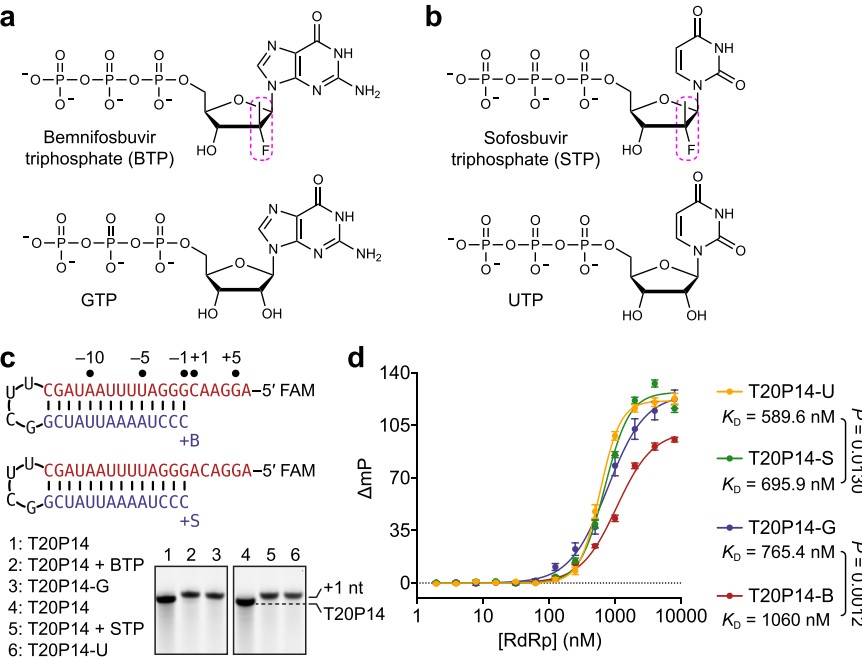

**Fig. 1 | Incorporation of bemnifosbuvir and sofosbuvir by SARS-CoV-2 RdRp.** Chemical structure of **a** bemnifosbuvir 5′-triphosphate (BTP) and **b** sofosbuvir 5′-triphosphate (STP) in comparison with their cognate natural NTPs. The characteristic 2′-fluoro-2′-C-methyl groups of bemnifosbuvir and sofosbuvir are highlighted by dashed magenta boxes. **c** Incorporation of BMP and SMP, respectively, into T20P14 RNAs by SARS-CoV-2 RdRp. Sequence and numbering of the substrate T20P14 RNAs used for bemnifosbuvir or sofosbuvir incorporation are indicated. T-RNA and P-RNA regions are colored in red and blue, respectively, and connected

by a UUCG tetraloop. **d** Fluorescence polarization analysis of the binding between SARS-CoV-2 RdRp and T20P14-U, T20P14-S, T20P14-G, or T20P14-B. Dissociation constant ($K_D$) values are indicated. Each data point represents the mean of nine biological replicates ± SEM. Dissociation constant ($K_D$) values are indicated. Statistical analyses were performed using the two-sided extra sum-of-squares F test. $P$ values for the comparisons of $K_D$ values are indicated. Source data are provided as a Source Data file.

been investigated for their potential use to combat COVID-19. While they exhibit inhibitory effects on SARS-CoV-2 RdRp and viral replication in vitro[29,37], their antiviral potencies against SARS-CoV-2, particularly that of sofosbuvir, are markedly lower than their potencies against HCV[26,28,40,41]. The undermined anti-SARS-CoV-2 effectiveness of these two NAs is, at least partially, due to the SARS-CoV-2 proofreading ExoN[42]. Indeed, incorporated bemnifosbuvir 5′-monophosphate (BMP) and sofosbuvir 5′-monophosphate (SMP) are subjected to ExoN excision, although at relatively lower rates than that of other nucleotide analogs, such as remdesivir 5′-monophosphate[15,29]. To evaluate the resistance of SARS-CoV-2 to bemnifosbuvir and sofosbuvir and determine the molecular underpinnings of such resistance, we first prepared and purified BMP- or SMP-terminated RNA by incubating a hairpin RNA that comprises a 20-nucleotide (nt) template strand (T-RNA) region and a 14-nt product strand (P-RNA) region (hereafter referred to as T20P14) (Fig. 1c) with purified SARS-CoV-2 RdRp in the presence of bemnifosbuvir 5′-triphosphate (BTP) or sofosbuvir 5′-triphosphate (STP). Polyacrylamide gel electrophoresis (PAGE) revealed that both NAs were readily and fully incorporated into the hairpin T20P14 RNA (the resulting RNAs are designated T20P14-B and T20P14-S, respectively) (Fig. 1c). Fluorescence polarization analyses showed that BMP- or SMP-terminated RNA exhibited lower RdRp-binding affinity than their respective standard RNA counterparts (designated T20P14-G and T20P14-U, respectively) did (Fig. 1d). The weakened RdRp-binding capability of T20P14-B and T20P14-S is likely due to the substitution of a favorable bi-furcated hydrogen bond formed between the 2′-OH group of a standard 3′-end NMP and S759 of nsp12, the core subunit of the RdRp, with a weaker fluoro-hydroxyl hydrogen bond[43,44] mediated by the 2′-fluoro group of either BMP or SMP (Supplementary Fig. 5). Therefore, our results suggest that incorporation of bemnifosbuvir or sofosbuvir into viral RNA destabilizes the RdRp-RNA complex and may trigger a dissociation of the NA-terminated RNA from RdRp.

## ExoN rescues BMP- or SMP-inhibited RNA synthesis

To evaluate how the 3′-end incorporated bemnifosbuvir or sofosbuvir affect the recognition of RNA by the coronavirus proofreading ExoN, we measured the binding affinities of T20P14-B or T20P14-S to SARS-CoV-2 ExoN and found that the 3′-end BMP or SMP strengthens the binding of the RNAs to ExoN compared with RNAs ended with the natural nucleotide GMP or UMP (Fig. 2a). The results indicate that the incorporation of BMP or SMP facilitates the transfer of the NA-terminated RNAs from RdRp to ExoN.

To assess the excision of RNA 3′-end BMP or SMP by SARS-CoV-2 ExoN, we determined the time course of ExoN digestion of the T20P14-B and T20P14-S in comparison with T20P14-G and T20P14-U, respectively (Fig. 2b, c). Our results showed that although all four RNAs were cleaved by ExoN, the digestion rate of BMP- or SMP-terminated RNA was noticeably lower than that of GMP- or UMP-ended RNA (Fig. 2b, c), suggesting that the modifications on the ribose rings of BMP and SMP likely hinder the rapid excision of the two NAs from RNA by ExoN. To determine how ExoN may undermine the inhibition of RdRp by bemnifosbuvir and sofosbuvir, hence the anti-coronavirus effectiveness of these two NAs, we examined the RNA extension from T20P14-B or T20P14-S in the absence or presence of ExoN (Fig. 2d, e). Our results showed that the incorporation of bemnifosbuvir and sofosbuvir almost abolished or largely blocked further extension of the RNA (Fig. 2d, e). However, a considerable portion of the bemnifosbuvir- or sofosbuvir-inhibited RNA extension was rescued when wild-type (WT) ExoN was also present in the reaction mixture. By contrast, such rescue effects were not observed when the ExoN contained a catalytically inactivating mutation E191A (Fig. 2d, e). Furthermore, to evaluate the impact of RdRp on ExoN-mediated excision of 3′-end BMP or SMP, we compared the digestion rates of T20P14-B or T20P14-S in the absence or presence of RdRp and found that RdRp enhances the excision of

both nucleotide analogs from the 3′-end of RNA by ExoN (Supplementary Fig. 6). Taken together, these results indicate that, despite moderate resilience of bemnifosbuvir and sofosbuvir to ExoN excision after being incorporated into RNAs, their RdRp inhibitory potency and likely their anti-coronavirus effectiveness are still greatly compromised by the viral proofreading ExoN.

## Structural basis of BMP and SMP recognition by ExoN

To elucidate the molecular basis of BMP and SMP excision by ExoN and reveal the structural determinants that confer the moderate level of resilience of the two NAs to ExoN excision, we assembled two complexes of SARS-CoV-2 ExoN E191A with either T20P14-B or T20P14-S and then performed single-particle cryo-EM analyses of the two ExoN•RNA complexes (designated ExoN•T20P14-B complex and ExoN•T20P14-S complex, respectively) (Supplementary Figs. 1–4). Similar to that observed in the previous structural study of the SARS-CoV-2 ExoN•standard RNA complex[45], the complexes of ExoN with the two NA-incorporated RNAs also exist in multiple different oligomeric states. The ExoN•T20P14-B complex predominantly adopts a dimeric form, with a small amount of the complex in monomeric form, in the cryo-EM dataset, whereas the ExoN•T20P14-S particles are mostly distributed between a tetrameric and monomeric form. (Supplementary Figs. 1c, 2c, and 3b). The final maps for the three structures were refined to an overall resolution of 2.9 Å, 2.4 Å, and 2.6 Å, respectively (Fig. 3a, b and Supplementary Fig. 7a). In the dimeric form of the ExoN•T20P14-B complex, the residues surrounding the ExoN active site of each protomer are not contacted by the other protomer (Supplementary Fig. 7b). Therefore, the active site conformation of ExoN and recognition of RNA 3′-end BMP is unlikely affected by the dimerization. By contrast, each protomer in the tetrameric form of the ExoN•T20P14-S complex interacts with two adjacent protomers through two distinct interfaces (Supplementary Fig. 7c, d). The interaction between protomer A and protomer B is mediated by the ExoN domain of nsp14 (Supplementary Fig. 7c), whereas protomer A contacts protomer C mainly through residues in the guanine N7 methyltransferase (N7-MTase) domain of nsp14 (Supplementary Fig. 7d). Despite these inter-protomer interactions, the overall structures and the ExoN active site conformations of the tetrameric and monomeric forms of the ExoN•T20P14-S complex are highly similar (Supplementary Fig. 7e, f), indicating the tetramerization does not affect the active site conformation of and substrate recognition by ExoN. A similar observation was also made in the comparison of different oligomeric forms of the ExoN•standard RNA complex[45]. Unless otherwise stated, we use the tetrameric form of ExoN•T20P14-S complex for subsequent structural analyses of sofosbuvir recognition by ExoN due to its higher resolution, hence better ability to clearly reveal the atomic-level details of the interactions between ExoN and the NA.

While the overall structures of the ExoN•T20P14-B and ExoN•T20P14-S are highly similar to that of the ExoN•standard RNA complex (Supplementary Fig. 7g, h), structural superimpositions revealed striking changes in the ExoN active site conformations of the BMP- and SMP-containing complexes. In the active sites of the ExoN•T20P14-B complexes, although the scissile phosphate between the 3′-end BMP (−1$B_P$) and −1$C_P$ is coordinated (Fig. 3c) just like that observed in the ExoN•standard RNA complexes[45], the critical catalytic residue H268 undergoes a major shift (>9 Å) away from the scissile phosphate due to a major structural rearrangement of the nsp14 α4-α5 loop harboring this residue (Fig. 3d), likely resulting in a substantially reduced activity of ExoN. As to the ExoN•T20P14-S complex, the underlying particles used to reconstruct both the monomeric and tetrameric structures appear to be heterogeneous in terms of the conformation of the nsp14 α4-α5 loop, reflected in the cryo-EM map showing densities corresponding to both the inactive conformation (Fig. 3e and Supplementary Fig. 7i, colored in orange, Fig. 3f and Supplementary Fig. 7j) and the activated conformation (Fig. 3e and

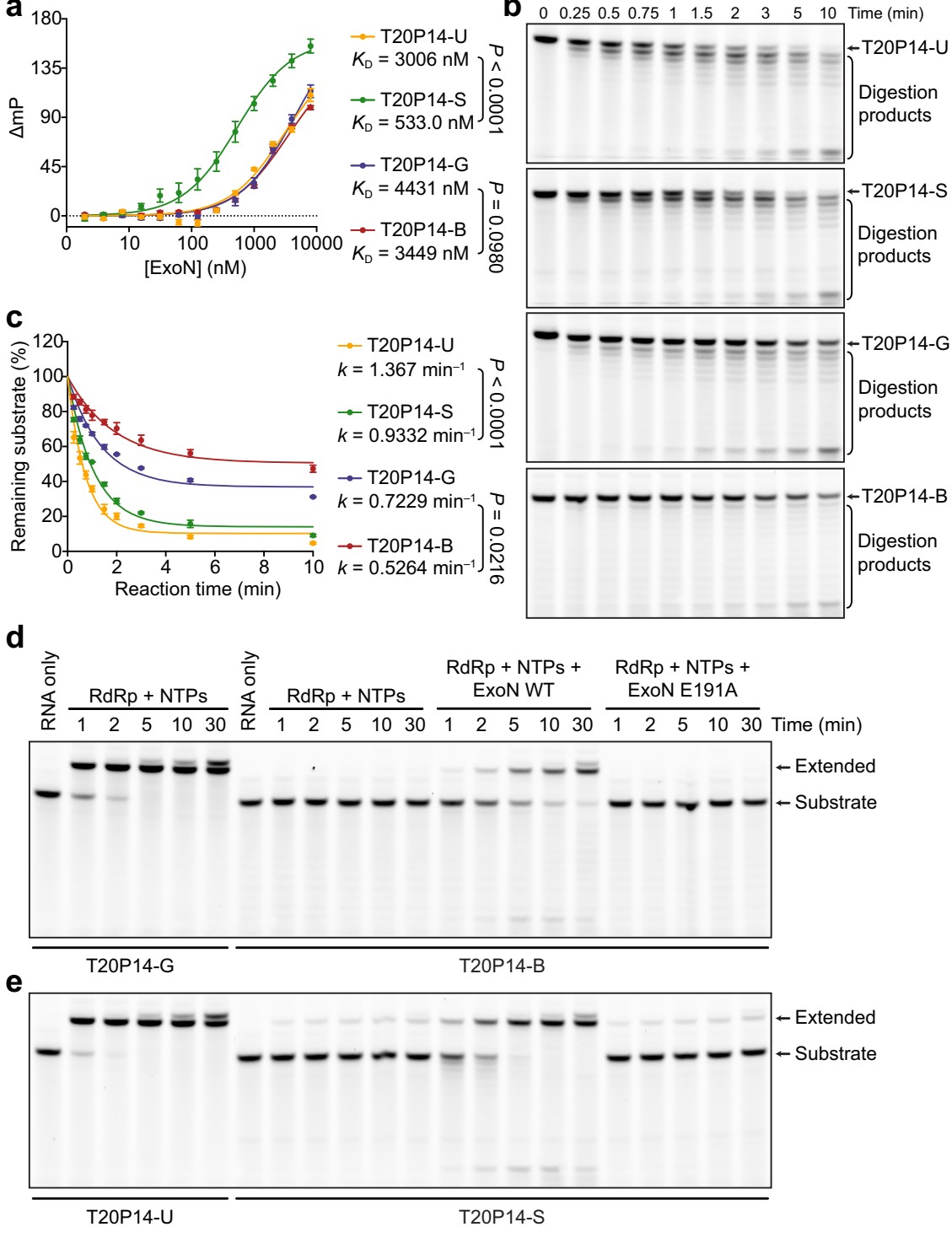

**Fig. 2 | Excision of bemnifosbuvir and sofosbuvir by SARS-CoV-2 ExoN.**
**a** Fluorescence polarization analysis of the binding between SARS-CoV-2 ExoN and T20P14-U, T20P14-S, T20P14-G, or T20P14-B. Each data point represents the mean of nine biological replicates ±SEM. Dissociation constant ($K_D$) values are indicated. Statistical analyses were performed using the two-sided extra sum-of-squares F test. $P$ values for the comparisons of $K_D$ values are indicated. **b** Digestion of T20P14-U, T20P14-S, T20P14-G, or T20P14-B by SARS-CoV-2 ExoN. The exonucleolytic digestion reactions were stopped at the indicated time points. The RNA products were resolved by denaturing PAGE and visualized by FAM imaging. A representative result from three biological replicates is shown. **c** Percentages of substrate RNAs remaining shown in (**b**) were quantified using Bio-Rad Image Lab from three independent experiments and are shown as mean ± SEM. The results were plotted in GraphPad Prism using the One-phase decay model. Decay rates ($k$) of T20P14-U, T20P14-S, T20P14-G, and T20P14-B are indicated. Statistical analyses were performed using the two-sided extra sum-of-squares F test. $P$ values for the comparisons of decay rate constants are indicated. SARS-CoV-2 ExoN rescues **d** bemnifosbuvir- or **e** sofosbuvir-inhibited RNA synthesis. RNA extension reactions were performed in the absence or presence of WT ExoN or ExoN E191A mutant and stopped at the indicated time points. The RNA products were resolved by denaturing PAGE and visualized by FAM imaging. A representative result from three biological replicates is shown. Source data are provided as a Source Data file.

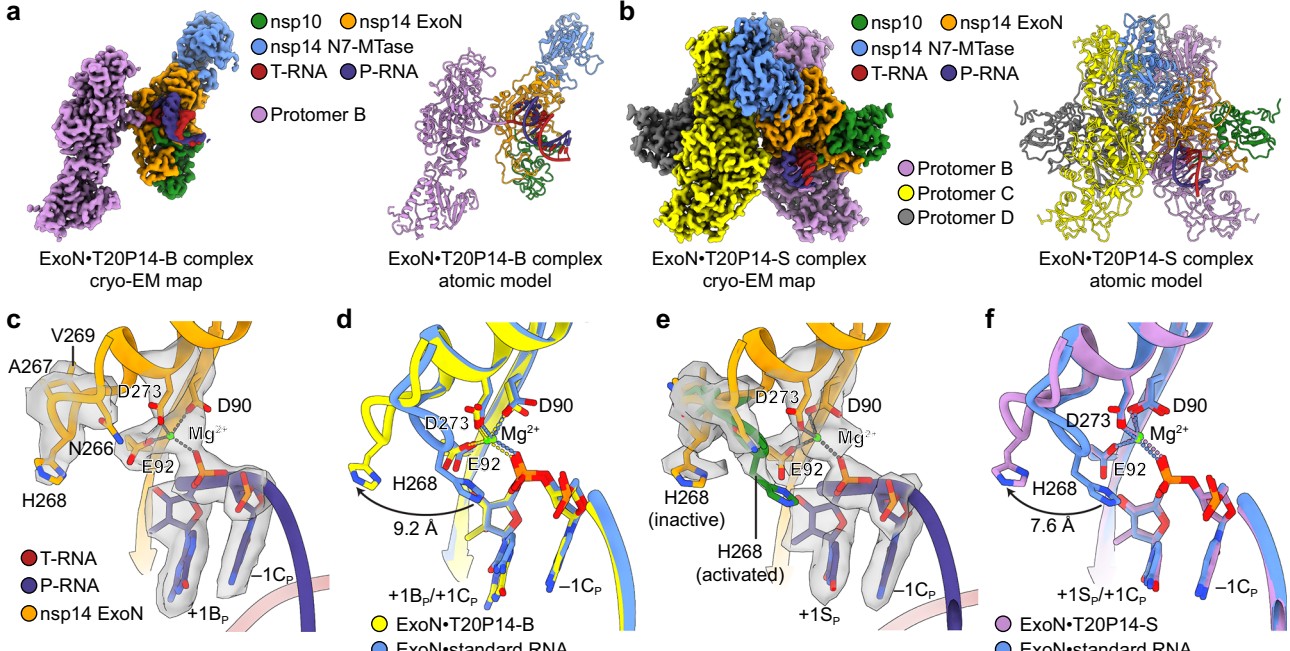

**Fig. 3 | Structural basis of recognition of bemnifosbuvir and sofosbuvir by SARS-CoV-2 ExoN.** a Cryo-EM map and atomic model of SARS-CoV-2 ExoN E191A mutant in complex with T20P14-B RNA (referred to as ExoN•T20P14-B complex). b Cryo-EM map and atomic model of SARS-CoV-2 ExoN E191A mutant in complex with T20P14-S RNA (referred to as ExoN•T20P14-S complex) (tetrameric form). c ExoN active site conformation of the ExoN•T20P14-B complex. Metal coordination bonds are shown as gray dashed lines. $+1B_P$, $-1C_P$, $Mg^{2+}$ ion, three catalytic carboxylate residues, and nsp14 α4-α5 loop are superimposed with their cryo-EM densities. d Superimposition of ExoN active site in the presence of T20P14-B or a standard RNA (PDB 7N0C). Root-mean-square deviation (RMSD) of the

superimposition is indicated. The major shift of H268 upon the binding of T20P14-B is indicated. e ExoN active site conformation of the ExoN•T20P14-S complex (tetrameric form). $+1S_P$, $-1C_P$, $Mg^{2+}$ ion, three catalytic carboxylate residues, and nsp14 α4-α5 loop are superimposed with their cryo-EM densities. The inactive and activated conformations of the α4-α5 loop are colored in orange and green, respectively. f Superimposition of ExoN active site in the presence of T20P14-S (tetrameric form) or a standard RNA (PDB ID 7N0D). For clarity, only the inactive conformation of the nsp14 α4-α5 loop in the ExoN•T20P14-S complex is shown. RMSD of the superimposition is indicated.

Supplementary Fig. 7i, colored in green). Such displacements of H268 from the ExoN catalytic center in both the ExoN•T20P14-B and ExoN•T20P14-S complexes are even more dramatic than that in the ExoN apo structure (Supplementary Fig. 7k, l), in which H268 is not in close proximity to other catalytic residues to support catalysis[45,46]. The tendency of ExoN to adopt an inactive conformation upon the binding of a BMP- or SMP-terminated RNA explains the moderate level of resistance to ExoN excision of these two incorporated NAs.

**Allosteric regulation of ExoN activity**

To delineate the structural basis of the ExoN-deactivating capabilities of BMP and SMP, we next examined their interactions with ExoN, particularly those involving the ribose rings of the NAs. While the hydrogen bonds mediated by the terminal ribose 3′-OH are maintained in the ExoN•T20P14-B and ExoN•T20P14-S complexes (Fig. 4a, b), the favorable interactions formed between ExoN residues and the ribose 2′-OH of standard cytidine 5′-monophosphate (CMP)[45] are lost. In place of the missing 2′-OH group, the 2′-CH₃ group of BMP and SMP inserts into a hydrophobic pocket formed by nsp14 P141 and F146 and establishes new interactions with the two residues (Fig. 4a, b). These additional hydrophobic interactions may contribute to the higher ExoN-binding affinities of T20P14-B and T20P14-S than their standard RNA counterparts (Fig. 2a) and, at the same time, also perturb the structure of the nsp14 P140–L149 loop, in which P141 and F146 reside, upon the binding of either of these two NA-containing RNAs.

The P140–L149 loop is situated opposite the H268-harboring α4-α5 loop at the bottom of the RNA-binding pocket (Fig. 4c–e). In the ExoN•T20P14-B complex structure, the P140–L149 loop turns -2–3 Å further away from the ExoN active site and scissile phosphate

compared with that in the ExoN•standard RNA complex structure (Fig. 4c). When the ExoN active site adopts a catalytically competent conformation with the binding of a standard RNA substrate, Q145 and H148 in the P140–L149 loop establish two hydrogen bonds with the side chain and main chain of the catalytic residue H268, respectively (Fig. 4d). These interactions are expected to stabilize H268 and help hold the H268-harboring α4-α5 loop in the activated conformation. In the presence of the BMP-terminated RNA, however, the shift of the P140–L149 loop widens the gap between this structural element and the α4-α5 loop, breaks the above two stabilizing hydrogen bonds, and leaves H268 to switch to an inactivated conformation (Fig. 4e). Such deprivation of interactions between the two loop structures is reminiscent of that observed in the ExoN apo structure (Supplementary Fig. 8a), whose active site is therefore not fully assembled[45,46]. In the ExoN•T20P14-S complex structure, the shift of the P140–L149 loop is more modest (Supplementary Fig. 8b) but is sufficient to greatly weaken its bridging interactions with the α4-α5 loop (Supplementary Fig. 8c, d) and drive a significant portion of the underlying complex particles to adopt the inactive state (Fig. 3e). To determine the contributions of residues in the P140–L149 loop to the excision of 3′-end BMP and SMP by ExoN, we measured and compared the digestion rates of the two NA-incorporated RNAs by either wild-type (WT) ExoN or ExoN carrying P141A, Q145A, F146A, or H148A mutation (Fig. 4f, g). Our exoribonuclease assay results show that all four mutations greatly impair the digestion of T20P14-S RNA (Fig. 4g). As to the digestion of T20P14-B RNA, whereas the Q145A mutation does not significantly affect the ExoN-mediated excision of the 3′-end BMP, likely because Q145 has a minimum contribution to the interaction between the P140–L149 loop and the inactive conformation of H268 in the ExoN•T20P14-B complex (Fig. 4e), mutating each of the other three

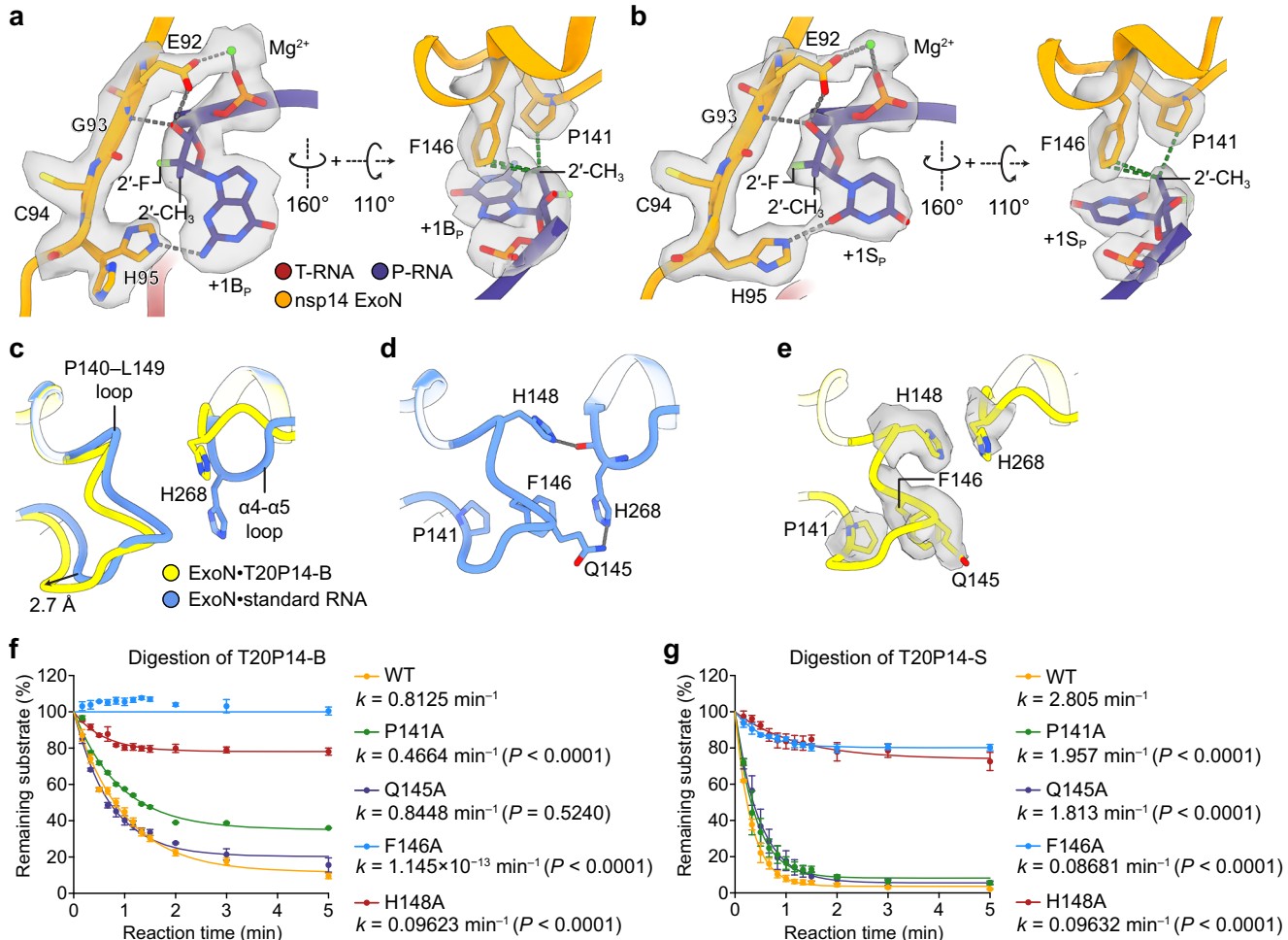

**Fig. 4 | Allosteric regulation of ExoN activity and excision of bemnifosbuvir and sofosbuvir. a** Detailed interactions between SARS-CoV-2 ExoN and RNA 3′-end BMP. Metal coordination bonds and hydrogen bonds are shown as gray dashed lines. Hydrophobic interactions are shown as green dashed lines. $Mg^{2+}$ ions, green spheres. $+1B_P$, $Mg^{2+}$ ion, and nsp14 residues are superimposed with their cryo-EM densities. **b** Detailed interactions between SARS-CoV-2 ExoN and RNA 3′-end SMP. Metal coordination bonds and hydrogen bonds are shown as gray dashed lines. Hydrophobic interactions are shown as green dashed lines. $Mg^{2+}$ ions, green spheres. $+1S_P$, $Mg^{2+}$ ion, and nsp14 residues are superimposed with their cryo-EM densities. **c** Superimposition of nsp14 P140–L149 loop and α4-α5 loop between ExoN•T20P14-B and ExoN•standard RNA (PDB 7N0C) complexes. The displacement of the P140–L149 loop upon the binding of T20P14-B RNA is indicated. **d** Bridging interactions between the nsp14 P140–L149 loop and α4-α5 loop in the ExoN•standard RNA complex structure. Hydrogen bonds are shown as gray dashed lines. **e** Loss of bridging interactions between the nsp14 P140–L149 loop and α4-α5 loop in the ExoN•T20P14-B complex structure. Nsp14 residues are superimposed with their cryo-EM densities. An important role of the nsp14 P140–L149 loop in the excision of **f** 3′-end BMP and **g** 3′-end SMP by ExoN. Exonucleolytic digestion of T20P14-B or T20P14-S RNA by WT or mutant forms of ExoN was stopped at indicated time points, and RNA products were resolved by denaturing PAGE and visualized by FAM imaging. The percentage of substrate RNA remaining at each time point was quantified using Bio-Rad Image Lab from three independent experiments and is shown as mean ± SEM. Rate constant ($k$) values are determined using the One-phase decay model in GraphPad Prism. Statistical analyses were performed using the two-sided extra sum-of-squares F test. $P$ values for the comparisons of decay rate constants between WT and each mutant are indicated. Source data are provided as a Source Data file.

residues substantially undermines or completely abolishes the RNA digestion (Fig. 4f).

To further interrogate the role of the P140–L149 loop in ExoN activity towards standard RNA substrates, we systematically examined the digestion of a series of RNAs bearing a 3′-end adenosine (designated T20P14-A), uridine (T20P14-U), cytidine (T20P14-C), or guanosine (T20P14-G) by either WT or the above four mutant forms (P141A, Q145A, F146A, or H148A) of ExoN. Our results show that all four mutants exhibit markedly lower ExoN activity in digesting each of the four standard RNA substrates (Fig. 5a–d). On one hand, the disruptive effects of Q145A and H148A mutants on ExoN-mediated RNA digestion further support the critical roles of these two residues in ExoN activity, likely through stabilizing the activated conformation of H268 as suggested by our structure analyses (Fig. 4d). On the other hand, the impaired ExoN activity caused by P141A or F146A mutation is

consistent with our structural observation that P141 and F146 are pivotal for a proper recognition of the 3′-end nucleotide[45] or NA (Fig. 4a, b), hence the optimal binding of the RNA substrate to ExoN. Notably, P141, Q145, F146, and H148 are highly conserved across all four genera of coronaviruses (Fig. 5e). Therefore, our findings reveal the nsp14 P140–L149 loop as a conserved allosteric regulator of coronavirus ExoN catalytic activity that senses different types of RNA substrates to finely tune the assembly of the ExoN active site and full activation of the enzyme.

### Structural basis of nucleotide preference of ExoN
Our structures of ExoN in complex with an RNA containing either 3′-BMP or SMP provide a unique opportunity to understand how ExoN recognizes different nucleotides in its catalytic center, which is essential for its role as an RNA proofreader in removing different types

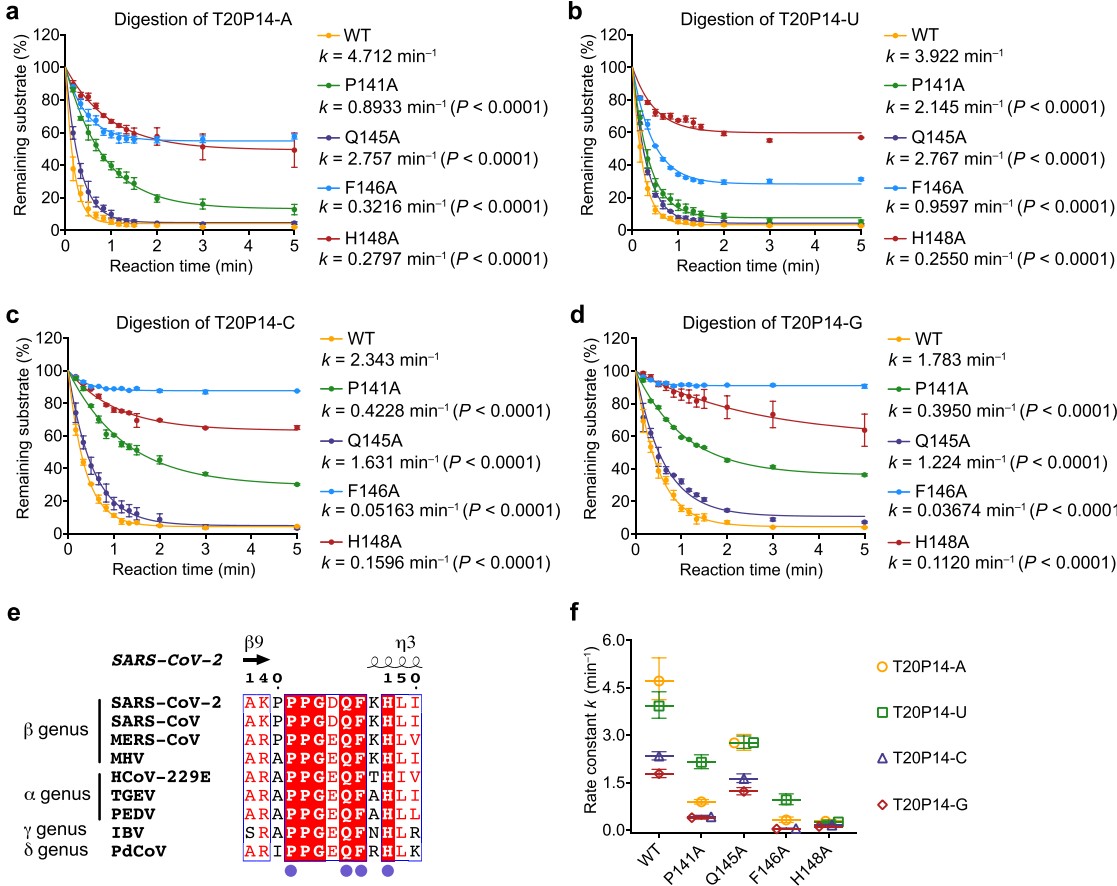

**Fig. 5 | Contribution of nsp14 P140–L149 loop to enzymatic activity and substrate preference of ExoN.** Exonucleolytic digestion of **a** T20P14-A, **b** T20P14-U, **c** T20P14-C, or **d** T20P14-G RNA by WT or mutant forms of ExoN was stopped at indicated time points, and RNA products were resolved by denaturing PAGE and visualized by FAM imaging. The percentage of substrate RNA remaining at each time point was quantified using Bio-Rad Image Lab from three independent experiments and is shown as mean ± SEM. Rate constant ($k$) values are determined using the One-phase decay model in GraphPad Prism. Statistical analyses were performed using the two-sided extra sum-of-squares F test. $P$ values for the comparisons of decay rate constants between WT and each mutant are indicated. **e** Sequence alignment of nsp14 from different coronaviruses. The residues in the P140-L149 loop that are involved in the allosteric regulation of ExoN active site assembly and tested in the mutagenesis analyses are completely conserved and indicated as purple dots. **f** Comparison of the rate constants ($k$) of RNA digestion by WT or mutant forms of ExoN. Data are shown as best-fit rate constant values ± 95% confidence interval (CI) of three biological replicates determined from curve fitting in (**a–d**). Source data are provided as a Source Data file.

of mis-incorporated nucleotides from viral RNAs. Specific recognition of different bases of the RNA 3′-end nucleotide is at least partially conferred by nsp14 H95 (Supplementary Fig. 9a–c), which forms a hydrogen bond with the cytosine base in the ExoN•standard RNA complex[45] and with the uracil base in the ExoN•T20P14-S complex (Fig. 4b and Supplementary Fig. 9a, b). As to the ExoN•T20P14-B complex, a hydrogen bond between H95 and the guanine base of BMP is maintained (Fig. 4a and Supplementary Fig. 9c). However, the side chain of H95 needs to switch to a much less preferred rotamer conformation (7.78% probability, according to the Rotamer Analysis tool in *Coot*[47]) than those observed when it interacts with the cytosine or uracil base (an average of 36.72% probability between the two structures) (Supplementary Fig. 9c). Moreover, the cryo-EM map of ExoN•T20P14-B complex shows an alternative conformation of H95 side chain, which points away from and does not contact with the 3′-end BMP (Fig. 4a), further indicating the less favorable recognition of a guanine base by ExoN. These structural observations explain the markedly lower ExoN-binding affinities and lower ExoN digestion rate of GMP- or BMP-terminated RNA than those of RNAs containing 3′-end UMP or SMP (Fig. 2a, c). Another probable contributing factor to the lower digestion rate of BMP- or GMP-terminated RNAs is the higher energy penalty of breaking the terminal $C_T$:$G_P$ base pair, a prerequisite for positioning the 3′-end guanine base in the ExoN active site for

excision, than the separation of a weaker base pair or mismatched pair. Indeed, our exoribonuclease assay shows that WT ExoN is most active in excising a 3′-end adenosine 5′-monophosphate (AMP) or uridine 5′-monophosphate (UMP) while least active in digesting a 3′-end GMP (Fig. 5f).

Interestingly, mutating P141 or F146 in the nsp14 P140–L149 loop alters the substrate preference of ExoN. While the P141A and F146A mutant forms of ExoN remain more active in digesting RNA substrate with a weak 3′-end base-pair ($A_T$:$U_P$ or $U_T$:$A_P$) than digesting RNA with a strong 3′-end base-pair ($G_T$:$C_P$ or $C_T$:$G_P$), they have a reversed order of preference for 3′-end AMP and UMP compared with WT ExoN (Fig. 5f). Although there lacks a structure of ExoN in complex with an AMP-terminated RNA, a comparative analysis of the ExoN-T20P14-B and ExoN-T20P14-S complexes reveals that nsp14 P141 and F146 make more extensive interactions with a 3′-end purine base than with a pyrimidine base (Supplementary Fig. 9d, e). Such different interaction patterns explain the more severe impact on the digestion of 3′-end AMP than the digestion of 3′-end UMP when P141 and F146 are mutated (Fig. 5f).

## Discussion

While NA-based antivirals work wonders in combating many DNA and RNA viruses[17–21,48,49], they have not seen much success in treating

coronavirus diseases. One major cause underlying the suboptimal effectiveness of NA-based antivirals against coronaviruses is the unique proofreading mechanism provided by the coronavirus ExoN that removes the NAs from viral RNAs despite their incorporation by the viral RdRp[14,29,36,37]. However, how the proofreading ExoN of coronaviruses recognizes and excises NA-based antivirals from viral RNAs, thus compromising their anti-coronavirus potencies, has been a long-standing puzzle in the field.

In this study, we determined the cryo-EM structures of SARS-CoV-2 ExoN in complex with RNA substrates containing two different incorporated NAs. Combined with comprehensive analyses of RdRp-RNA and ExoN-RNA interactions and exoribonucleolytic digestion of RNAs, we reveal that the incorporation of bemnifosbuvir or sofosbuvir both facilitates the transfer of the NA-containing RNAs from the RdRp active site to the ExoN active site, which may be a common theme for NA-based RdRp inhibitors. The diminished RdRp-RNA interaction likely results from less favorable contacts mediated by the modified chemical groups on the NAs, whereas the tighter bindings of NA-terminated RNAs to ExoN are explained by the additional favorable interactions formed between ExoN and these NAs compared with their respective cognate natural nucleotides. However, the increased ExoN-binding affinities of the NA-terminated RNAs do not always translate to higher digestion rates of these RNAs by ExoN. In fact, RNA substrates with 3′-end BMP or SMP exhibit moderate levels of resistance to ExoN excision. In-depth structural analyses of the ExoN•RNA complex containing BMP or SMP reveal an unexpected conformational change of the ExoN active site, likely induced by the newly established interactions between the 2′-CH$_3$ group of the NAs and an allosteric regulatory loop (P140–L149) in ExoN. These interactions, which are otherwise absent when ExoN binds to a standard RNA, likely displace the allosteric regulatory loop of nsp14 and destabilize the catalytically competent conformation of the ExoN active site, hence impairing the enzymatic activity of ExoN.

Furthermore, our cryo-EM structures of ExoN in complex with BMP and SMP, which are analogs of GMP and UMP, respectively, also offer important insights into the broad-spectrum recognition of different nucleotides by ExoN. The ability of the ExoN active site to accommodate different 3′-end nucleotides is, at least partially, dictated by the versatile hydrogen bonding patterns of nsp14 H95, which forms a specific hydrogen bond with the 3′-end uracil, guanine, or cytosine base in the ExoN•T20P14-S, ExoN•T20P14-B, or ExoN•standard RNA complex, respectively. However, as shown by our structures, the recognition of a 3′-end guanine base is much less preferred in the ExoN active site, explaining the greatly lower ExoN-binding affinities and ExoN-digestion rates of BMP- or GMP-terminated RNAs compared with other RNAs. Therefore, our findings suggest that NAs with a guanine-like base and the ability to perturb the allosteric regulatory loop of ExoN, thus preventing its active site assembly, would be good candidates for ExoN-resistant RdRp inhibitors.

While our manuscript is being finalized, another study (Wang et al.) reported the structure of SARS-CoV-2 ExoN in complex with an RNA substrate containing a terminal SMP or BMP, respectively[50]. However, the structural and biochemical characterizations in the Wang et al. study suggested that ExoN excises 3′-end BMP as efficiently as it removes a 3′-end natural nucleotide, which is contradictory to our results (Fig. 4b, c) and is not supported by previous findings[29,51]. Such a discrepancy is likely due to different RNA substrates used in our work [double-stranded RNA (dsRNA)] and the Wang et al. study [single-stranded RNA (ssRNA)]. Considering ExoN is not active in digesting a homopolymeric ssRNA[36], the ExoN-mediated digestion of the BMP- or SMP-terminated ssRNA observed in the Wang et al. study likely occurred after the formation of RNA secondary structures. Such weakly formed secondary structures may be dynamic and deviate from the structure of native BMP- or SMP-terminated dsRNA substrate generated via the incorporation of the two NAs by RdRp during viral

RNA synthesis. In addition, because of the markedly higher resolution of our ExoN•T20P14-S (overall 2.4 Å with local resolution at the ExoN active site approaching 2.2 Å) and ExoN•T20P14-B (overall 2.9 Å with local resolution at the ExoN active site approaching 2.2 Å) complexes than those reported by the Wang et al. study (overall resolutions of 4.2 Å and 3.1 Å for the two structures, respectively)[50], we were able to unambiguously build the 3′-end BMP and SMP into their respective cryo-EM densities and reveal (i) the additional key interactions established between the ribose 2′-CH$_3$ group of SMP or BMP and P141 and F146, (ii) the resulting disruption of the stabilizing interactions between the allosteric regulatory loop (P140–L149) and the H268-harboring α4-α5 loop, and (iii) the unfavorable accommodation of the guanine base of 3′-end BMP by nsp14 H95 at the bottom of RNA-binding pocket. These novel structural observations were not made by the Wang et al. study[50] but nicely explained the varied levels of resistance to ExoN excision of BMP and SMP as revealed by our (Fig. 2b, c) and previous biochemical results[29,51].

Taken together, our study reveals the mechanisms of ExoN-mediated coronavirus resistance to multiple NAs that are among the most important anti-coronavirus drugs in clinical use or trials. Additionally, the insights gained from our results will guide the development of new RdRp-targeting NAs that can overcome the ExoN-mediated RNA proofreading mechanism.

## Methods

### Protein expression and purification

The full-length genes of SARS-CoV-2 nsp7, nsp8, and nsp12 (GenBank accession number NC_045512.2) were chemically synthesized with codon optimization for expression in *Escherichia coli* (*E. coli*) (GENE-WIZ). The −1 ribosomal frameshifting that naturally occurs in the virus to produce nsp12 was corrected in the synthesized gene. The genes were fused to an N-terminal His$_6$-Smt3 tag and were cloned into a pET21a vector (Millipore Sigma) between NdeI and XhoI restrictive sites. The genes of SARS-CoV-2 nsp10 were chemically synthesized with codon optimization for expression in (*E. coli*) (Integrated DNA Technologies). The gene of SARS-CoV-2 nsp14 was requested from Addgene. The SARS-CoV-2 nsp14 genes were cloned into a pETDuet-1 vector with an N-terminal His$_6$-Smt3 tag between the NcoI and HindIII restrictive sites. The SARS-CoV-2 nsp10 genes were cloned into a pETDuet-1 vector between the NdeI and XhoI restrictive sites. P141A, Q145A, F146A, H148A, or E191A mutation of nsp14 was introduced by site-directed mutagenesis. The mutation was confirmed by Sanger sequencing. Plasmids are available upon request.

All proteins were overexpressed in *E. coli* BL21 Star (DE3) (ThermoFisher Scientific) at 17 °C for 18 h. Cells were resuspended in buffer A (50 mM 4-(2-hydroxyethyl)-1-piperazine ethanesulfonic acid (HEPES), pH 7.5, 200 mM NaCl, 5% glycerol, 1 mM β-mercaptoethanol (β-ME), 20 mM imidazole) and lysed using a sonicator (QSonica). The cell lysate was cleared by centrifugation at 19,500 rpm using a JA-25.50 rotor (Beckman Coulter) for 1 h at 4 °C. The clarified cell lysate was loaded onto a HisTrap HP affinity chromatography column (Cytiva Life Sciences) and eluted through a linear gradient from 100% buffer A to 40% buffer A mixed with 60% buffer B (50 mM HEPES, pH 7.5, 500 mM NaCl, 5% glycerol, 1 mM β-ME, 500 mM imidazole). Eluted protein samples were loaded onto a HiTrap Heparin HP column (Cytiva Life Sciences) and eluted with buffer C (20 mM HEPES, pH 7.0, 1 M NaCl, 2 mM β-ME), followed by overnight incubation with Ulp1 protease to remove the N-terminal His$_6$-Smt3 tag. All proteins were subsequently purified by size-exclusion chromatography (SEC) on a HiLoad 16/600 Superdex 200 pg column (Cytiva Life Sciences) in buffer D (20 mM HEPES, pH 7.5, 300 mM NaCl, 4 mM MgCl$_2$, 1 mM Tris(2-carboxyethyl) phosphine hydrochloride (TCEP-HCl)). ExoN complex was assembled by mixing nsp10 and nsp14 in a 1:1 molar ratio at room temperature for 30 min. RdRp complex was assembled by mixing nsp12, nsp7, and nsp8 in a 1:1:2 molar ratio at room temperature for 30 min.

### In vitro transcription and RNA purification

The DNA templates for in vitro transcription of T20P14 RNAs were generated by annealing two DNA oligonucleotides with complementary sequences (5′- CAG TAA TAC GAC TCA CTA TAG GGA ACG GGA TTT TAA TAG CTT CGG CTA TTA AAA TCC C -3′ and 5′-[G_m][G_m]G ATT TTA ATA GCC GAA GCT ATT AAA ATC CCG TTC CCT ATA GTG AGT CGT ATT ACT G-3′ for subsequent generation of T20P14-B RNAs; 5′- CAG TAA TAC GAC TCA CTA TAG GGA CAG GGA TTT TAA TAG CTT CGG CTA TTA AAA TCC C -3′ and 5′-[G_m][G_m]G ATT TTA ATA GCC GAA GCT ATT AAA ATC CCT GTC CCT ATA GTG AGT CGT ATT ACT G-3′ for the subsequent generation of T20P14-S RNAs) in a 1:1 molar ratio. T7 RNA polymerase (RNAP) φ6.5 promoter sequence in the non-template DNA strands is underlined. Two nucleotides denoted [G_m] on the 5′ end of the template DNA strand are 2′-O-methylated to improve the 3′-end homogeneity of the RNA transcripts[52].

In vitro transcription reaction (5 mL) was assembled using 2 nmol annealed DNA template, 500 µg T7 RNAP and 200 µl 25× RNAsecure RNase inactivation reagent (ThermoFisher Scientific) in 1× reaction buffer (80 mM HEPES, pH 7.5, 24 mM MgCl₂, 40 mM DTT, 2 mM spermidine, 4 mM of each NTP) and incubated at 37 °C for 2 h. The reaction mixture was centrifugation at 4000 × g for 10 min at 4 °C to remove pyrophosphate precipitate and subsequently quenched by adding ethylenediaminetetraacetic acid (EDTA) to a final concentration of 50 mM. The RNA transcripts were extracted with phenol:chloroform:isoamyl alcohol (25:24:1) (ThermoFisher Scientific) three times, followed by purification on a Sephadex G-25 PD-10 desalting column (Cytiva Life Sciences) and a HiLoad 16/600 Superdex 200 pg SEC column in buffer E (10 mM HEPES, pH 7.0, 50 mM NaCl).

5′-FAM-labeled RNAs are chemically synthesized (MilliporeSigma), followed by purification on a Superdex 200 Increase 10/300 GL SEC column (Cytiva Life Sciences) in buffer E.

To generate nucleotide analog-incorporated RNAs, purified T20P14 RNAs were incubated with pre-assembled SARS-CoV-2 RdRp and 0.3 mM of bemnifosbuvir triphosphate (MedChemExpress) or sofosbuvir triphosphate (MedChemExpress) at 37 °C for 2 h in buffer F (25 mM HEPES, pH 7.5, 75 mM NaCl, 4 mM MgCl₂, and 1 mM TCEP). The reaction mixture was centrifugation at 4000 × g for 10 min at 4 °C to remove pyrophosphate precipitate and subsequently quenched by adding EDTA to a final concentration of 50 mM. The RNA transcripts were extracted with phenol:chloroform:isoamyl alcohol (25:24:1) three times to completely remove the RdRp, followed by purification on a Superdex 200 Increase 10/300 GL SEC column in buffer E.

The sequences, modifications, and sources of all RNAs used in this study are reported in Supplementary Table 2.

### Fluorescence polarization assays

FAM-labeled T20P14 series of RNAs at a final concentration of 6 nM were incubated with a 2x serial dilution series (ranging from 1.953 nM to 8 µM) of pre-assembled SARS-CoV-2 RdRp WT or ExoN E191A mutant complex at room temperature in buffer G (25 mM HEPES, pH 7.5, 50 mM NaCl, 4 mM MgCl₂, and 1 mM TCEP) in a 384-well plate. Fluorescence polarization was measured on a Victor Nivo multimode microplate reader (Revvity) with the excitation and emission wavelengths of 480 nm and 530 nm, respectively. Changes in fluorescence polarization ($\Delta mP$) upon protein binding were plotted against RdRp WT or ExoN E191A concentration in GraphPad Prism. The data for ExoN•RNA binding were fitted using a custom "One site-specific binding with ligand depletion" model[53] $[Y = B_{max} \times (X − F \times Y/B_{max})/(K_D + X − F \times Y/B_{max})$, where $X$ is the total protein concentration, $F$ is the total fluorescence probe concentration, $Y$ is the change of fluorescence polarization from the RNA-only control group, $B_{max}$ is maximum binding in the same units as $Y$, and $K_D$ is the dissociation constant in the same unit as $X$] to determine the $K_D$ values and 95% confidence intervals. The data for RdRp•RNA binding were fitted using a custom "Specific binding with variable slope and ligand depletion" model

$[Y = B_{max} \times (X − F \times Y/B_{max})^n/(K_D{}^n + (X − F \times Y/B_{max})^n)$, where n is the Hill slope] to determine the $K_D$ values and 95% confidence intervals. Statistical analyses for comparing the best-fit $K_D$ values between each group were performed using the two-sided extra sum-of-squares F test.

### Exoribonuclease assays

WT SARS-CoV-2 nsp10•nsp14 complex at a final concentration of 20 nM was incubated with FAM-T20P14-G, FAM-T20P14-B, FAM-T20P14-U, or FAM-T20P14-S at a final concentration of 3 µM at 37 °C in buffer G. The reactions were stopped at 15 s, 30 s, 45 s, 60 s, 90 s, 120 s, 3 min, 5 min, and 10 min, respectively, by adding an equal volume of 2 × TBE-Urea sample buffer supplemented with 50 mM EDTA and heating at 75 °C for 5 min. To compare the digestion rate of FAM-T20P14-B or FAM-T20P14-S in the absence versus in the presence of RdRp, WT SARS-CoV-2 nsp10•nsp14 complex at a final concentration of 10 nM was incubated with each of the two NA-incorporated RNAs at a final concentration of 1 µM at 37 °C in buffer G. The reactions were stopped at 15 s, 30 s, 45 s, 60 s, 75 s, 90 s, 120 s, 3 min, 5 min, and 10 min, respectively. To determine the contribution of P141, Q145, F146, or H148 to ExoN activity and substrate preference, WT, P141A, Q145A, F146A, or H148A mutant form of SARS-CoV-2 nsp10•nsp14 complex each at a final concentration of 20 nM was incubated with FAM-T20P14-A, FAM-T20P14-C, FAM-T20P14-G, FAM-T20P14-B, FAM-T20P14-U, or FAM-T20P14-S each at a final concentration of 1 µM at 37 °C in buffer G. The reactions were stopped at 10 s, 20 s, 30 s, 40 s, 50 s, 60 s, 70 s, 80 s, 90 s, 120 s, 3 min, and 5 min, respectively, by adding an equal volume of 2 × TBE-Urea sample buffer supplemented with 50 mM EDTA and heating at 75 °C for 5 min. The cleavage products were resolved on denaturing 16% polyacrylamide gels and visualized by fluorescent imaging on a ChemiDoc MP imager (Bio-Rad). The RNA band corresponding to the substrate RNA at each reaction time point was quantified using Image Lab Software Suite (Bio-Rad). Percentages of substrate RNAs remaining were plotted against their respective reaction times in GraphPad Prism. The results were subjected to curve-fitting using the One-phase decay model to determine the rate constant ($k$) of RNA digestion for each reaction. Statistical analyses for comparing the best-fit rate constant ($k$) values between each group were performed using the two-sided extra sum-of-squares F test.

### Rescue of stalled RNA synthesis assay

FAM-labeled T20P14 series of RNAs at a final concentration of 1 µM were incubated with pre-assembled SARS-CoV-2 RdRp at a final concentration of 1.25 µM at 30 °C for 5 min in buffer F. The reactions were started by adding NTPs, a mixture of NTPs and 10 nM SARS-CoV-2 nsp10•nsp14 WT, or a mixture of NTPs and 10 nM SARS-CoV-2 nsp10•nsp14 E191A mutant. NTPs were supplied at a final concentration of 0.2 mM each. The reactions were stopped at different time points by adding an equal volume of 2 × TBE-Urea sample buffer supplemented with 50 mM EDTA and heating at 75 °C for 5 min. RNA products were resolved on denaturing 16% polyacrylamide gels and visualized by fluorescent imaging on a ChemiDoc MP imager (Bio-Rad). The RNA band corresponding to the substrate RNA at each reaction time point was quantified using Image Lab Software Suite (Bio-Rad).

### Complex assembly of SARS-CoV-2 ExoN with NA-incorporated RNAs

The two SARS-CoV-2 ExoN•RNA complexes in this study were reconstituted by mixing SARS-CoV-2 nsp10•nsp14 E191A mutant complex with T20P14-B or T20P14-S, respectively, in a 1:2 molar ratio and incubating the mixture at 30 °C for 30 min in buffer H (25 mM HEPES, pH 7.5, 200 mM NaCl, 4 mM MgCl₂, and 1 mM TCEP). The assembled SARS-CoV-2 ExoN•T20P14-B complex was purified using a Superdex 200 Increase 10/300 GL SEC column in buffer H. The chromatography

fractions corresponding to the ExoN•T20P14-B complex were collected for subsequent single-particle cryo-EM analysis. For the ExoN•T20P14-S complex, the assembled complex was either purified using a Superdex 200 Increase 10/300 GL SEC column in buffer H or directly used for cryo-EM sample preparation without SEC purification.

### Cryo-EM sample preparation and data acquisition

Purified SARS-CoV-2 ExoN•RNA complexes (A260 = 3) were mixed with 8 mM of 3-([3-Cholamidopropyl]dimethylammonio)-2-hydroxy-1-propanesulfonate (CHAPSO) immediately before grid preparation. 3.5 μl of each complex was applied to freshly glow-discharged Quantifoil 300 mesh holey carbon grids with R1.2/1.3 hole pattern (Electron Microscopy Sciences). Grids were blotted for 5 s at 22 °C under 100% relative humidity and plunge-frozen in liquid nitrogen-cooled liquid ethane. The cryo-EM dataset for the SARS-CoV-2 ExoN•T20P14-B complex and the cryo-EM dataset 1 for the SARS-CoV-2 ExoN•T20P14-S complex were collected on a Titan Krios electron microscope (ThermoFisher Scientific) operated at 300 kV equipped with a BioQuantum K3 detector (Gatan, Inc.) at the Hormel Institute, University of Minnesota. For the dataset of ExoN•T20P14-B complex, the movie frames were collected at a nominal magnification of 81,000×, corresponding to 1.0724 Å per pixel, at a dose rate of 20.4 e⁻ per physical pixel per second, with a defocus range of −1.0 to −2.0 μm. The total exposure time for each movie is 3 s, thus resulting in a total accumulated dose of 53.33 e⁻/Å2, which was fractionated into 40 frames. For the dataset 1 of ExoN•T20P14-S complex, the movie frames were collected at a nominal magnification of 81,000×, corresponding to 1.0724 Å per pixel, at a dose rate of 20.37 e⁻ per physical pixel per second, with a defocus range of −1.0 to −2.0 μm. The total exposure time for each movie is 3 s, thus resulting in a total accumulated dose of 53.13 e⁻/Å2, which was fractionated into 40 frames. The cryo-EM dataset 2 of ExoN•T20P14-S complex was collected on a Titan Krios electron microscope operated at 300 kV equipped with a Falcon 4i detector and Selectris X imaging filter at the Stanford-SLAC Cryo-EM Center (S²C²). The movie frames were collected at a nominal magnification of 130,000 ×, corresponding to 0.9254 Å per pixel, at a dose rate of 7.829 e⁻ per physical pixel per second, with a defocus range of −1.0 to −2.0 μm. The total exposure time for each movie is 4.65 s, thus resulting in a total accumulated dose of 42.51 e⁻/Å2, which was fractionated into 40 frames. The statistics of cryo-EM data collection are summarized in Supplementary Table 1.

### Cryo-EM image processing

Dose-fractioned cryo-EM movies were imported into cryoSPARC[54] for patch-based motion correction and patch-based CTF estimation, followed by blob picking and Topaz picking[55]. The picked particles were subjected to three rounds of 2D classifications to remove junk particles. Particles in good 2D classes were selected for the generation of multiple ab initio models, which were subsequently low-pass filtered to 20 Å and used as starting references for heterogeneous refinement in cryoSPARC or global 3D classification in RELION-5.0[56,57]. The particle stacks corresponding to good classes resulting from global 3D classification or heterogeneous refinement were subjected to iterative rounds of CTF refinement[58], Reference-based motion correction[59,60], and non-uniform refinement[61] to generate the final cryo-EM map.

To improve the map quality and interpretability of the SARS-CoV-2 ExoN•T20P14-B complex, the final particle stacks corresponding to the complex were subjected to particle subtraction to retain only the signal of the protomer A of the dimeric ExoN•RNA complexes, followed by masked local 3D refinement in cryoSPARC. The final non-uniform refined maps for the three complexes were further improved by density modification using Phenix Resolve[62] without supplying a structural model to avoid model bias. The overall map resolution was calculated based on the Fourier shell correlation (FSC) cutoff at 0.143 between two half-maps, after applying a soft mask to exclude the bulk solvent region. The maps were sharpened automatically during non-uniform refinement or local refinement and post-processed using DeepEMhancer[63]. The raw maps, automatically sharpened maps, Resolve density-modified maps, and DeepEMhancer-processed maps were used as cross-references during model building. Local resolution estimation was calculated from the two half-maps in cryoSPARC and visualized in UCSF ChimeraX[64]. Histogram and direction FSC curves for cryo-EM maps were analyzed and generated by the Orientation Diagnosis tool[65,66] in cryoSPARC. As indicated by the orientation diagnosis and histograms of 3D FSC plots, the maps for the two complexes are free of preferred orientation issues.

### Cryo-EM model building and refinement

The cryo-EM structure of the monomeric form of SARS-CoV-2 ExoN•RNA complex (7N0C) was docked into the ExoN•T20P14-B complex map or the monomeric form of the ExoN•T20P14-S complex map as a rigid body. For the tetrameric form of the ExoN•T20P14-S complex, the tetrameric form of SARS-CoV-2 nsp10•nsp14•RNA complex (PDB 7N0D) was used for rigid-body docking. All docked models are then flexibility fitted[67] into each of the four ExoN•RNA complex maps. The protein and RNA subunits in each complex were manually rebuilt in Coot[47]. The resolution and density features of the cryo-EM maps are of sufficiently high quality for the unambiguous assignment of protein and RNA registers in each of the complexes. All atomic models were refined using Phenix real-space refinement[68] with secondary structure restraints, rotamer restraints, and Ramachandran restraints. The final structures were validated with MolProbity[69]. The statistics of cryo-EM refinement were summarized in Supplementary Table 1. Molecular representations were prepared using UCSF ChimeraX.

### Reporting summary

Further information on research design is available in the Nature Portfolio Reporting Summary linked to this article.

## Data availability

Atomic coordinates of the four structures determined in this study have been deposited in the Protein Data Bank with accession codes 9YRK (SARS-CoV-2 ExoN•T20P14-B complex, dimeric form), 9YRL (SARS-CoV-2 ExoN•T20P14-B complex, protomer A), 9YRN (SARS-CoV-2 ExoN•T20P14-S complex, tetrameric form), and 9YRO (SARS-CoV-2 ExoN•T20P14-S complex, monomeric form). The cryo-EM maps have been deposited in the Electron Microscopy Data Bank with accession numbers EMD-73369 (SARS-CoV-2 ExoN•T20P14-B complex, dimeric form), EMD-73370 (SARS-CoV-2 ExoN•T20P14-B complex, protomer A focus-refined map), EMD-73371 (SARS-CoV-2 ExoN•T20P14-S complex, tetrameric form), and EMD-73372 (SARS-CoV-2 ExoN•T20P14-S complex, monomeric form). Source data are provided with this paper.

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

## Acknowledgements

We thank P. Juneja for support with cryo-EM sample screening at the Iowa State University cryo-EM facility, cryo-EM facility staff at the Hormel Institute and G. Nye at the Stanford-SLAC Cryo-EM Center (S$^2$C2) for support during the collection of cryo-EM datasets. The S$^2$C2 is supported by a National Institutes of Health grant R24GM154186. This work was supported by a National Institutes of Health grant DP2AI177906 to C.L., an award from the Searle Scholars Program SSP-2024-106 to C.L., a grant from the Hormel Institute, University of Minnesota, to B.L., and a National Institutes of Health grant R35GM150607 to Y.Y.

## Author contributions

Y.Y. and C.L. conceived and designed the experiments. C.L., Y.Y., Y.L., X.C., and R.J.G. performed protein and RNA purifications. C.L., Y.L., and Y.Y. conducted biochemical characterizations. Y.Y. and C.L. prepared the cryo-EM samples. B.L. and C.L. collected cryo-EM data. C.L. and Y.Y. processed the cryo-EM data and performed model building and structural analyses. Y.Y. and C.L. wrote the manuscript with input from B.L.

## Competing interests

The authors declare no competing interests.
