## [Transparent Peer Review File · Nature Communications]

Mechanism of SARS-CoV-2 resistance to nucleotide analog-based antivirals

Corresponding Author: Dr Chang Liu

Version 0:

Reviewer comments:

Reviewer #1

(Remarks to the Author)

The authors elucidate the mechanism of SARS-CoV-2 resistance to 2 nucleotide analog-based antivirals (Bemnifosbuvir and Sofosbuvir) through a detailed structural analysis of interactions of NA-terminated RNAs with the ExoN enzyme.

The authors have done an excellent job and a thoughtful analysis. The paper is very clearly written, the review of existing literature is solid, and the discussion concise.

The authors have determined with an excellent degree of precision the determinants of excision in relation to which base is present at the 3'-end, with specific contacts established by H95. They have identified a loop which is key in the expression of activity, as well as a 9A shift of a key residue (H268) when activity is engaged.

Overall, the paper will be of great importance to refine our understanding of the NA-excision process, an essential step in designing new generation NAs active against CoVs.

Minor comments:

Line 57 : ...as AT-527), and sofosbuvir, have been shown to inhibit SARS-CoV-2 RdRp activity in vitro and, to some extent, viral replication in cells²²⁻²⁹.

Perhaps mention that Sofosbuvir activity, as reported in Sacramento et al., is very weak, to say the least, as opposed to HCV, for which it is very potent.

Line 58 : These nucleotide analogs are mis-incorporated into the viral RNAs and act as chain terminators to inhibit RdRp and thus stall RNA synthesis^{23,27,29}.

They are incorporated, not mis-incorporated. Change/refine wording.

Reviewer #2

(Remarks to the Author)

In this manuscript the authors investigate the mechanism of how SARS-CoV-2 Nsp14 counteracts nucleotide analog based antivirals. The authors demonstrated that the RdRp can incorporate both BTP and STP on short hairpins but that these nucleotide analogs weaken RdRp – RNA binding. Next they used in vitro assays to establish the Nsp14 can degrade the BTP and STP containing transcripts, however not as well as those with a natural nucleotide. Despite this weak activity the authors were able to show that Nsp14 can rescue nucleotide analog inhibited RNA extension through its proofreading activity. Finally, the authors determined two cryo-EM structures of the Nsp14-Nsp10 complex bound to both the BTP and STP containing RNA substrates. These structures revealed how the nucleoside analogues impair Nsp14 activity, and they provide additional information about Nsp14 specificity. Overall, this manuscript reveals high resolution snapshots of Nsp14-10 bound to two different nucleotide analogs and contributes to our understanding of how coronaviruses are able to resist NA-based antivirals.

Comments:

1. Is Nsp14 more efficient at proofreading in the presence of the RdRp? Could this explain why the authors do not observe

much exonuclease activity in Fig. 2b but still observe RNA synthesis in 2d?

2. I am perplexed by the different stoichiometries observed by Cryo-EM and the authors do not adequately describe this observation in the manuscript. Are the monomer, dimer, and tetramer forms all active? Is it known what the stoichiometry of Nsp10-Nsp14 is in virus infected cells? Why does the RNA substrate alter the distribution of these oligomeric forms? The SEC peak for the S-complex is much broader which would suggest it has a greater distribution of oligomeric species. The authors should provide additional information in the manuscript about these oligomeric species. Are the dimer/tetramer interfaces near the active site. Could the oligomeric state influence the observed active site conformations?

3. The authors propose an allosteric mode of regulation via the P140-L149 loop but there is no validation of this model. The authors should design a series of mutants to this regulatory loop and determine how these mutants impact activity (or cite previous studies that have mutated this loop). Given that the authors also suggest that guanine-like bases perturb the allosteric regulatory loop it would be interesting if specific point mutants alter the specificity of Nsp14. Are there any known Nsp14 variants in this region of the protein?

4. The authors results do not agree with work published earlier this year from Wang et al (JACS, 2025), in terms of Nsp14 activity against nucleoside analogue containing substrates. One key difference between these manuscripts is the RNA substrate. Wang et al used a ssRNA, as opposed to a hairpin. The authors need to consider the possibility that the RNA substrate may have a significant impact on Nsp14 activity not just the 3' nucleoside analogue.

Minor Comments:

- Fig. 3 Legend – The authors need to mention that this is an ExoN mutant.
- RdRp vs Nsp12 nomenclature. The authors primarily use RdRp throughout the manuscript but refer to it as Nsp12 in a few places.

Version 1:

Reviewer comments:

Reviewer #2

(Remarks to the Author)

The authors have addressed all of my previous concerns and I am fully supportive of publication.

Response to reviewers

We thank the reviewers for their many helpful suggestions and comments. Addressing these points has allowed us to strengthen the manuscript through the addition of new data and clarifications in the text. Below, we provide a detailed response in blue to all reviewers' comments.

Reviewer #1 (Remarks to the Author):

The authors elucidate the mechanism of SARS-CoV-2 resistance to 2 nucleotide analog-based antivirals (Bemnifosbuvir and Sofosbuvir) through a detailed structural analysis of interactions of NA-terminated RNAs with the ExoN enzyme.

The authors have done an excellent job and a thoughtful analysis. The paper is very clearly written, the review of existing literature is solid, and the discussion concise.

The authors have determined with an excellent degree of precision the determinants of excision in relation to which base is present at the 3'-end, with specific contacts established by H95. They have identified a loop which is key in the expression of activity, as well as a 9A shift of a key residue (H268) when activity is engaged.

Overall, the paper will be of great importance to refine our understanding of the NA-excision process, an essential step in designing new generation NAs active against CoVs.

Response: We very much appreciate the reviewer's enthusiastic comment about the significance of our study for antiviral drug development and the quality and presentation of our results.

Minor comments:

Line 57 : ...as AT-527), and sofosbuvir, have been shown to inhibit SARS-CoV-2 RdRp activity in vitro and, to some extent, viral replication in cells²²⁻²⁹.

Perhaps mention that Sofosbuvir activity, as reported in Sacramento et al., is very weak, to say the least, as opposed to HCV, for which it is very potent.

Response: We appreciated the reviewer's point. We have added the following sentence in the revised manuscript (Lines 99–101):

“While they exhibit inhibitory effects on SARS-CoV-2 RdRp and viral replication in vitro^{29,37}, their antiviral potencies against SARS-CoV-2, particularly that of sofosbuvir, are markedly lower than their potencies against HCV^{26,28,40,41}.”

Line 58 : These nucleotide analogs are mis-incorporated into the viral RNAs and act as chain terminators to inhibit RdRp and thus stall RNA synthesis^{23,27,29}.

They are incorporated, not mis-incorporated. Change/refine wording.

Response: As suggested by the reviewer, we have changed “mis-incorporated” to “incorporated” in this sentence.

Reviewer #2 (Remarks to the Author):

In this manuscript the authors investigate the mechanism of how SARS-CoV-2 Nsp14 counteracts nucleotide analog based antivirals. The authors demonstrated that the RdRp can incorporate both BTP and STP on short hairpins but that these nucleotide analogs weaken RdRp – RNA binding. Next they used in vitro assays to establish the Nsp14 can degrade the BTP and STP containing transcripts, however not as well as those with a natural nucleotide. Despite this weak activity the authors were able to show that Nsp14 can rescue nucleotide analog inhibited RNA extension through its proofreading activity. Finally, the authors determined two cryo-EM structures of the Nsp14-Nsp10 complex bound to both the BTP and STP containing RNA substrates. These structures revealed how the nucleoside analogues impair Nsp14 activity, and they provide additional information about Nsp14 specificity. Overall, this manuscript reveals high resolution snapshots of Nsp14-10 bound to two different nucleotide analogs and contributes to our understanding of how coronaviruses are able to resist NA-based antivirals.

Response: We thank the reviewer for the concise summary of our work and the positive comments on the significance and impact of our study.

Comments:

1. Is Nsp14 more efficient at proofreading in the presence of the RdRp? Could this explain why the authors do not observe much exonuclease activity in Fig. 2b but still observe RNA synthesis in 2d?

Response: We thank the reviewer for this intriguing question. In the revised manuscript, we have performed new exoribonuclease assays and compared the ExoN-mediated excision of bemnifosbuvir and sofosbuvir in the absence or presence of RdRp. Our result showed that RdRp enhances the excision of the two nucleotide analogs from the 3' end of RNA by ExoN.

We have incorporated the new result as Supplementary Fig. 6 and added the following text in the revised manuscript (Lines 143–146):

“Furthermore, to evaluate the impact of RdRp on ExoN-mediated excision of 3'-end BMP or SMP, we compared the digestion rates of T20P14-B or T20P14-S in the absence or presence

of RdRp and found that RdRp enhances the excision of both nucleotide analogs from the 3'-end of RNA by ExoN (Supplementary Fig. 6)."

2. I am perplexed by the different stoichiometries observed by Cryo-EM and the authors do not adequately describe this observation in the manuscript. Are the monomer, dimer, and tetramer forms all active? Is it known what the stoichiometry of Nsp10-Nsp14 is in virus infected cells? Why does the RNA substrate alter the distribution of these oligomeric forms? The SEC peak for the S-complex is much broader which would suggest it has a greater distribution of oligomeric species. The authors should provide additional information in the manuscript about these oligomeric species. Are the dimer/tetramer interfaces near the active site. Could the oligomeric state influence the observed active site conformations?

Response: We thank the reviewer for raising these important points. In the revised manuscript, we have collected a new cryo-EM dataset of the ExoN•T20P14-S complex and reconstructed a cryo-EM map of the monomeric form of this complex at a resolution of 2.6 Å (Supplementary Fig. 3) (Although the original cryo-EM dataset of the ExoN•T20P14-S complex also contained considerable amount of monomeric particles, we were unable to obtain a high-resolution map of this complex form, likely due to the limited contrast of the micrographs in the original dataset). The structural superimposition of the monomeric and tetrameric forms of the ExoN•T20P14-S complex shows that both the overall structure and ExoN active site conformation of the two oligomeric forms of ExoN•T20P14-S are almost identical, suggesting that tetramerization does not affect the active site conformation of and substrate recognition by the ExoN. As to the ExoN•T20P14-B complex, the residues surrounding the ExoN active site of each protomer are not contacted by the other protomer (the shortest distance is >7.5 Å). Therefore, the active site conformation of ExoN is unlikely to be affected by the dimerization.

To our knowledge, there has been no study examining the oligomeric state of the nsp10-nsp14 in virus-infected cells. However, the repeated observation of different oligomeric forms in the cryo-EM datasets of multiple different nsp10-nsp14-RNA complexes (including the ExoN•T20P14-S and ExoN•T20P14-B complexes reported in the current study and the ExoN•standard RNA complex in our previous study (PMID: 34315827)) assembled in physiologically relevant conditions suggests that some or all of these oligomeric forms may also exist in virus-infected cells. The broader peak of the ExoN•T20P14-S complex than that of the ExoN•T20P14-B complex on their respective SEC purification chromatograms is mostly because of the more prominent difference in the molecular sizes between the tetrameric form and monomeric form. The molecular size difference between the dimeric form and monomeric form of the ExoN•RNA complex is not substantial enough to allow adequate separation of the two oligomeric states on the SEC column. While the exact mechanism underlying the influence of different substrates on the distribution of nsp10-nsp14 oligomeric forms is a great topic of future investigation, we hypothesize that the binding of different RNA substrates may allosterically change the conformations of the

oligomerization interfaces. These structural rearrangements, although subtle, can alter the free energies of the formation of different oligomers and lead to a considerable redistribution of these oligomeric forms.

As suggested by this reviewer, we have added the following text in the revised manuscript to provide more information about these oligomeric species of the ExoN•RNA complexes (Lines 157–179):

“Similar to that observed in the previous structural study of the SARS-CoV-2 ExoN•standard RNA complex⁴⁵, the complexes of ExoN with the two NA-incorporated RNAs also exist in multiple different oligomeric states. The ExoN•T20P14-B complex predominantly adopts a dimeric form, with a small amount of the complex in monomeric form, in the cryo-EM dataset, whereas the ExoN•T20P14-S particles are mostly distributed between a tetrameric and monomeric form. (Supplementary Figs. 1c, 2c, and 3b). The final maps for the three structures were refined to an overall resolution of 2.9 Å, 2.4 Å, and 2.6 Å, respectively (Fig. 3a, b and Supplementary Fig. 7a). In the dimeric form of the ExoN•T20P14-B complex, the residues surrounding the ExoN active site of each protomer are not contacted by the other protomer (Supplementary Fig. 7b). Therefore, the active site conformation of ExoN and recognition of RNA 3'-end BMP is unlikely affected by the dimerization. By contrast, each protomer in the tetrameric form of the ExoN•T20P14-S complex interacts with two adjacent protomers through two distinct interfaces (Supplementary Fig. 7c, d). The interaction between protomer A and protomer B is mediated by the ExoN domain of nsp14 (Supplementary Fig. 7c), whereas protomer A contacts protomer C mainly through residues in the guanine N7 methyltransferase (N7-MTase) domain of nsp14 (Supplementary Fig. 7d). Despite these inter-protomer interactions, the overall structures and the ExoN active site conformations of the tetrameric and monomeric forms of the ExoN•T20P14-S complex are highly similar (Supplementary Fig. 7e, f), indicating the tetramerization does not affect the active site conformation of and substrate recognition by ExoN. A similar observation was also made in the comparison of different oligomeric forms of the ExoN•standard RNA complex⁴⁵. Unless otherwise stated, we use the tetrameric form of ExoN•T20P14-S complex for subsequent structural analyses of sofosbuvir recognition by ExoN due to its higher resolution, hence better ability to clearly reveal the atomic-level details of the interactions between ExoN and the NA.”

3. The authors propose an allosteric mode of regulation via the P140-L149 loop but there is no validation of this model. The authors should design a series of mutants to this regulatory loop and determine how these mutants impact activity (or cite previous studies that have mutated this loop). Given that the authors also suggest that guanine-like bases perturb the allosteric regulatory loop it would be interesting if specific point mutants alter the specificity of Nsp14. Are there any known Nsp14 variants in this region of the protein?

Response: We appreciate the reviewer's constructive suggestion for examining the impact of mutations in the P140-L149 loop on ExoN activity. In the revised manuscript, we have constructed and purified four mutants (P141A, Q145A, F146A, and H148A). Among these four residues that are mutated, P141 and F146 interact with the 2'-CH₃ group of BMP and SMP as well as the nucleobases of the two NAs and the 3'-end standard nucleotide. Q145 and H148 are proposed to be involved in stabilizing the catalytic residue H268 in its activated conformation, hence promoting the full assembly of the ExoN active site. Our exoribonuclease assays performed with these mutants, in comparison with WT ExoN, revealed that the mutation of all the above four residues significantly reduces ExoN activity towards RNAs ending with either the two nucleotide analogs or four standard nucleotides. This biochemical validation supports the important role of the P140-L149 loop in ExoN activity.

Intriguingly, our results also revealed that P141A and F146A mutations alter the substrate preference of ExoN. While GMP remains as the least preferred nucleotide at 3' end, mostly because of the higher free energy penalty of breaking the terminal C:G base pair and the unfavorable recognition of the guanine base by nsp14 H95, for all the four tested mutants, P141A and F146A mutants exhibit much higher excision activity towards a 3'-end UMP than towards a 3'-end AMP, which is a reversed order of preference compared with that of WT ExoN. Structural analyses of the interactions between P141 and F146 of nsp14 and different 3'-end nucleobases reveal different patterns of recognition of a purine base versus a pyrimidine base (Supplementary Fig. 9d, e), explaining the more severe impact on the digestion of 3'-end AMP than the digestion of 3'-end UMP when P141 and F146 are mutated.

Regarding the nsp14 variants, according to PMID 36811085 and UniProt annotation (P0DTD1), no nsp14 variant identified to date shows mutations of the P140-L149 loop, further suggesting an essential role of this loop in ExoN activity.

In the revised manuscript, we have incorporated the new results as Fig. 4f, g and Fig. 5 and added the following text (Lines 227–251 and Lines 273–286):

“To determine the contributions of residues in the P140–L149 loop to the excision of 3'-end BMP and SMP by ExoN, we measured and compared the digestion rates of the two NA-incorporated RNAs by either wild-type (WT) ExoN or ExoN carrying P141A, Q145A, F146A, or H148A mutation (Fig. 4f, g). Our exoribonuclease assay results show that all four mutations greatly impair the digestion of T20P14-S RNA (Fig. 4g). As to the digestion of T20P14-B RNA, whereas the Q145A mutation does not significantly affect the ExoN-mediated excision of the 3'-end BMP, likely because Q145 has a minimum contribution to the interaction between the P140–L149 loop and the inactive conformation of H268 in the ExoN•T20P14-B complex (Fig. 4e), mutating each of the other three residues substantially undermines or completely abolishes the RNA digestion (Fig. 4f).

To further interrogate the role of the P140–L149 loop in ExoN activity towards standard RNA substrates, we systematically examined the digestion of a series of RNAs bearing a 3'-end adenosine (designated T20P14-A), uridine (T20P14-U), cytidine (T20P14-C), or guanosine (T20P14-G) by either WT or the above four mutant forms (P141A, Q145A, F146A, or H148A) of ExoN. Our results show that all four mutants exhibit markedly lower ExoN activity in digesting each of the four standard RNA substrates (Fig. 5a–d). On one hand, the disruptive effects of Q145A and H148A mutants on ExoN-mediated RNA digestion further support the critical roles of these two residues in ExoN activity, likely through stabilizing the activated conformation of H268 as suggested by our structure analyses (Fig. 4d). On the other hand, the impaired ExoN activity caused by P141A or F146A mutation is consistent with our structural observation that P141 and F146 are pivotal for a proper recognition of the 3'-end nucleotide⁴⁵ or NA (Fig. 4a, b), hence the optimal binding of the RNA substrate to ExoN. Notably, P141, Q145, F146, and H148 are highly conserved across all four genera of coronaviruses (Fig. 5e). Therefore, our findings reveal the nsp14 P140–L149 loop as a conserved allosteric regulator of coronavirus ExoN catalytic activity that senses different types of RNA substrates to finely tune the assembly of the ExoN active site and full activation of the enzyme.”

“Indeed, our exoribonuclease assay shows that WT ExoN is most active in excising a 3'-end adenosine 5'-monophosphate (AMP) or uridine 5'-monophosphate (UMP) while least active in digesting a 3'-end GMP (Fig. 5f).

Interestingly, mutating P141 or F146 in the nsp14 P140–L149 loop alters the substrate preference of ExoN. While the P141A and F146A mutant forms of ExoN remain more active in digesting RNA substrate with a weak 3'-end base-pair ($A_T:U_P$ or $U_T:A_P$) than digesting RNA with a strong 3'-end base-pair ($G_T:C_P$ or $C_T:G_P$), they have a reversed order of preference for 3'-end AMP and UMP compared with WT ExoN (Fig. 5f). Although there lacks a structure of ExoN in complex with an AMP-terminated RNA, a comparative analysis of the ExoN-T20P14-B and ExoN-T20P14-S complexes reveals that nsp14 P141 and F146 make more extensive interactions with a 3'-end purine base than with a pyrimidine base (Supplementary Fig. 9d, e). Such different interaction patterns explain the more severe impact on the digestion of 3'-end AMP than the digestion of 3'-end UMP when P141 and F146 are mutated (Fig. 5f).”

4. The authors results do not agree with work published earlier this year from Wang et al (JACS, 2025), in terms of Nsp14 activity against nucleoside analogue containing substrates. One key difference between these manuscripts is the RNA substrate. Wang et al used a ssRNA, as opposed to a hairpin. The authors need to consider the possibility that the RNA substrate may have a significant impact on Nsp14 activity not just the 3' nucleoside analogue.

Response: We thank the reviewer for this insightful comment. Considering ExoN has been shown to be completely inactive in digesting a homopolymeric ssRNA (PMID 35165203), the ExoN-mediated digestion of the BMP- or SMP-terminated ssRNA observed in the Wang et al.

study was likely only possible upon the formation of RNA secondary structures. Indeed, secondary structure prediction of the ssRNA used in the Wang *et al.* study shows that the ssRNA likely folds into a short hairpin (Fig. R1). In addition, such weakly formed secondary structures may be dynamic and deviate from the structure of native BMP- or SMP-terminated dsRNA substrate generated via the incorporation of the two NAs by RdRp during viral RNA synthesis. By contrast, the RNA substrate used in our study has a much more stably formed double-stranded region, which likely better mimics the BMP- or SMP-incorporated dsRNA formed during SARS-CoV-2 RdRp-mediated RNA synthesis.

Figure R1. Predicted secondary structure of the 16-nt ssRNA used in the Wang *et al.* study.

In the revised manuscript, we have added the following text in the Discussion section (Lines 326–337):

*“While our manuscript is being finalized, another study (Wang *et al.*) reported the structure of SARS-CoV-2 ExoN in complex with an RNA substrate containing a terminal SMP or BMP, respectively⁵⁰. However, the structural and biochemical characterizations in the Wang *et al.* study suggested that ExoN excises 3'-end BMP as efficiently as it removes a 3'-end natural nucleotide, which is contradictory to our results (Fig. 4b, c) and is not supported by previous findings^{29,51}. Such a discrepancy is likely due to different RNA substrates used in our work [double-stranded RNA (dsRNA)] and the Wang *et al.* study [single-stranded RNA (ssRNA)]. Considering ExoN is not active in digesting a homopolymeric ssRNA³⁶, the ExoN-mediated digestion of the BMP- or SMP-terminated ssRNA observed in the Wang *et al.* study likely occurred after the formation of RNA secondary structures. Such weakly formed secondary structures may be dynamic and deviate from the structure of native BMP- or SMP-terminated dsRNA substrate generated via the incorporation of the two NAs by RdRp during viral RNA synthesis.”*

Minor Comments:

- Fig. 3 Legend – The authors need to mention that this is an ExoN mutant.

Response: We have added such information in the Fig. 3 legend:

“(a) Cryo-EM map and atomic model of SARS-CoV-2 ExoN E191A mutant in complex with T20P14-B RNA (referred to as ExoN•T20P14-B complex). (b) Cryo-EM map and atomic model of SARS-CoV-2 ExoN E191A mutant in complex with T20P14-S RNA (referred to as ExoN•T20P14-S complex)”

- RdRp vs Nsp12 nomenclature. The authors primarily use RdRp throughout the manuscript but refer to it as Nsp12 in a few places.

Response: We apologize for this confusion. Our intention was to use RdRp when referring to the functional assembly of the enzyme and use nsp12 when describing specific amino acid residues in this protein subunit. To improve the clarity, we have added the following text when first mentioning nsp12 in the revised manuscript:

“... S759 of nsp12, the core subunit of the RdRp, ...”